

# A rapid and accurate method for the detection of four aminoglycoside modifying enzyme drug resistance gene in clinical strains of *Escherichia coli* by a multiplex polymerase chain reaction

Yaoqiang Shi[1,2,*], Chao Li[1,2,*], Guangying Yang[1,2], Xueshan Xia[1,2], Xiaoqin Mao[3], Yue Fang[1,2], A-Mei Zhang[1,2] and Yuzhu Song[1,2]

[1] Faculty of Life Science and Technology, Kunming University of Science and Technology, Kunming, Yunnan, China
[2] Molecular Medicine Center of Yunnan Province, Kunming, Yunnan, China
[3] The First People's Hospital of Yunnan Province, Kunming, Yunnan, China
* These authors contributed equally to this work.

Corresponding authors
A-Mei Zhang, zam1980@yeah.net
Yuzhu Song,
yuzhusong@kmust.edu.cn

## ABSTRACT

**Background:** Antibiotics are highly effective drugs used in the treatment of infectious diseases. Aminoglycoside antibiotics are one of the most common antibiotics in the treatment of bacterial infections. However, the development of drug resistance against those medicines is becoming a serious concern.

**Aim:** This study aimed to develop an efficient, rapid, accurate, and sensitive detection method that is applicable for routine clinical use.

**Methods:** *Escherichia coli* was used as a model organism to develop a rapid, accurate, and reliable multiplex polymerase chain reaction (M-PCR) for the detection of four aminoglycoside modifying enzyme (AME) resistance genes *Aac(6′)-Ib*, *Aac(3)-II*, *Ant(3″)-Ia*, and *Aph(3′)-Ia*. M-PCR was used to detect the distribution of AME resistance genes in 237 clinical strains of *E. coli*. The results were verified by simplex polymerase chain reaction (S-PCR).

**Results:** Results of M-PCR and S-PCR showed that the detection rates of *Aac(6′)-Ib*, *Aac(3)-II*, *Ant(3″)-Ia*, and *Aph(3′)-Ia* were 32.7%, 59.2%, 23.5%, and 16.8%, respectively, in 237 clinical strains of *E. coli*. Compared with the traditional methods for detection and identification, the rapid and accurate M-PCR detection method was established to detect AME drug resistance genes. This technique can be used for the clinical detection as well as the surveillance and monitoring of the spread of those specific antibiotic resistance genes.

## INTRODUCTION

With the abuse and misuse of antibiotics, bacterial tolerance is becoming an increasingly serious concern (*Ferri et al., 2017*; *Levin-Reisman et al., 2017*), leading to the emergence of a series of drug-resistant bacteria (multi-drug-resistance (MDR), extensively-drug-resistant

(XDR), and pan-drug-resistant (PDR)) (*Healey et al., 2016*; *Planet, 2017*). MDR is defined as acquired non-susceptibility to at least one agent in three or more antimicrobial categories. XDR is defined as non-susceptibility to at least one agent in all but two or fewer antimicrobial categories. PDR is defined as non-susceptibility to all agents in all antimicrobial categories (*Magiorakos et al., 2012*). Aminoglycoside antibiotics (Amikacin, Gentamicin, Tobramycin, Kanamycin, Netilmicin, Streptomycin, and Neomycin) (*Doi, Wachino & Arakawa, 2016*; *Krause et al., 2016*; *Yadegar et al., 2009*) are mainly used to treat infections caused by aerobic Gram-negative bacteria, such as *Escherichia coli* and *Klebsiella pneumoniae*, and non-fermenters like *Pseudomonas aeruginosa* (*Jana & Deb, 2006*; *Magnet & Blanchard, 2005*; *Zavascki, Klee & Bulitta, 2017*). However, bacteria can easily develop tolerance to aminoglycoside antibiotics due to the production of aminoglycoside modifying enzymes (AMEs) (*Haidar et al., 2016*; *Khosravi, Jenabi & Montazeri, 2017*) and the rapid transmission of AME resistance genes in pathogenic bacteria (*Ramirez & Tolmasky, 2010*). The severity of antibiotic tolerance has become a global worldwide concern. The theme of World Health Day in 2011 was "Combat Drug Resistance: No action today, No cure tomorrow" (*Chellat, Raguz & Riedl, 2016*; *Tseng et al., 2012*), whereas that in 2018 was "Change Can't Wait. Our Time with Antibiotics is Running Out". In such a severe situation of drug resistance, there is an urgent need to identify the species of bacteria and their drug resistance genes accurately and quickly to guide clinical drug use (*Brossier et al., 2017*; *Mu et al., 2016*). This need led to the establishment of a rapid, accurate, and economical method to detect pathogens and their drug resistance as early as possible. Such a technique is helpful for the rational use of drugs in clinical practice, as well as of great clinical significance to control and shorten the course of the disease (*Laamiri et al., 2016*).

Although the traditional method for bacterial resistance identification is simple and economical, identification is completed in about 4–7 days (*Jami Al-Ahmadi & Zahmatkesh Roodsari, 2016*; *Phaneuf et al., 2013*). It includes several steps, such as bacterial culture, single colony isolation, colony morphology observation, biochemical identification, and serotype identification (*Panek, Frac & Bilinska-Wielgus, 2016*). The accuracy of this method is low, and errors easily occur (*Tuttle et al., 2011*). The main methods used to test drug sensitivity include the disk diffusion method, *E*-test, dilution method (agar and broth dilution method), and automatic instruments (*Biswas, 2016*; *Ghosh et al., 2015*). Such methods have the advantages of low cost, easy operation, and strong flexibility. However, they also feature some inevitable disadvantages, such as slow, empirical dependance. With the development of biological science and technology, many new biological technologies have entered people's lives. Interspecific and intraspecific conserved nucleic acid sequences have been discovered in succession (*Nagar & Hahsler, 2013*), and various bioinformatics and molecular biological techniques based on nucleic acid amplification have been used to identify pathogens; an example of such techniques is polymerase chain reaction (PCR), which has become the gold standard (*Tuttle et al., 2011*).

Multiplex polymerase chain reaction (M-PCR) (*Chamberlain et al., 1988*) usually means that two or more pairs of primers exist in one PCR reaction system at the same time; it can amplify multiple target bands at one time (*Henegariu et al., 1997*). Its reaction

principle, operation steps, and general composition of the required reagents are basically the same as those of simplex PCR (S-PCR). To date, M-PCR has been widely used in many fields, such as scientific research and disease diagnosis, especially in the simultaneous detection of pathogenic microorganisms, hereditary diseases, and oncogenes (*Azizi et al., 2016*; *Skodvin et al., 2017*). What's more, M-PCR can also be used in the simultaneous detection of multidrug resistance genes in pathogenic bacteria (*Hong et al., 2009*). Compared with traditional drug sensitivity test and S-PCR, M-PCR is fast, efficient, and able to timely guide the clinical antibiotic therapy (*Chavada & Maley, 2015*; *Lee et al., 2014*; *Park & Ricke, 2015*). M-PCR has higher efficiency than S-PCR, and the former can detect a variety of pathogenic bacteria or drug-resistant genes simultaneously in one reaction system (*Kim, Hwang & Kim, 2017*). At the same time, M-PCR can reduce any experimental errors that may occurs during the experiment.

Among the AME resistance genes, *Aac(6′)-Ib*, *Aac(3)-II*, *Ant(3″)-Ia*, and *Aph(3′)-Ia* are the most widely distributed (*Costello et al., 2019*; *Haldorsen et al., 2014*; *Nasiri et al., 2018*; *Odumosu, Adeniyi & Chandra, 2015*; *Ojdana et al., 2018*; *Vaziri et al., 2011*; *Xiao & Hu, 2012*). Given that the M-PCR can detect multiple genes simultaneously, we aimed to develop a M-PCR system for the detection of the four most widely spread AME genes. The reaction system was verified in 237 clinical *E. coli* strains.

## MATERIALS AND METHODS

### Bacterial strains and culture

The 237 clinical strains of *E. coli* used in this study were provided, isolated, cultured, and identified by the First People's Hospital of Yunnan Province. All the strains were cultured in Luria-Bertani (LB) liquid medium in a shaking incubator at 37 °C and 180 rpm for 12 h. The bacterial genome was extracted using the TIANamp genomic DNA kit following the manufacturer's protocol and then stored at −40 °C for further experiments.

### Search of drug resistance genes searching and design of primers

Four AME resistance genes *Aac(6′)-Ib*, *Ant(3″)-Ia*, *Aph(3′)-Ia*, and *Aac(3)-II* were downloaded from the Comprehensive Antibiotic Resistance Database (https://card.mcmaster.ca/). The primers were designed according to the conservative region and synthesized by TSINGKE Biological Technology. The primers sequences are shown in Table 1.

### Establishment of S-PCR reaction system

S-PCR was performed using 2X Tsingke Master Mix, which was purchased from Tsingke Biotech Co., Ltd. (Kunming, China). According to the manufacturer's protocol, the S-PCR reaction system containing 12.5 µL of 2X Tsingke Master Mix, 1 µL of primers (10 µM), and 1 µg of DNA template from each strain was added with nuclease-free water up to 25 µL. The reactions were performed in a GeneAmp PCR System 9700 (Thermo Fisher Scientific, Inc., Waltham, MA, USA) with the following amplification conditions: pre-denaturation at 95 °C for 5 min; 30 cycles of denaturation at 95 °C for 30 s, annealing at 57 °C, 59 °C, 56 °C, and 57 °C (*Aac(6′)-Ib, Ant(3″)-Ia, Aph(3′)-Ia,* and

**Table 1 Primers used in this study.**

| Gene | Primers | Sequence (5′–3′) | Product size (bp) | Annealing temperature (°C) |
|---|---|---|---|---|
| Aac(6′)-Ib | Aac(6′)-I-F | AAACCCCGCTTTCTCGTAGC | 112 | 57 |
| | Aac(6′)-I-R | AAACCCCGCTTTCTCGTAGC | | |
| Ant(3″)-Ia | Ant(3″)-F | CCGGTTCCTGAACAGGATC | 180 | 59 |
| | Ant(3″)-R | CCCAGTCGGCAGCGACATC | | |
| Aph(3′)-Ia | Aph(3′)-F | CAAGATGGATTGCACGCAGG | 317 | 56 |
| | Aph(3′)-R | TTCAGTGACAACGTCGAGCA | | |
| Aac(3)-II | Aac(3)-II-F | GCTCGGTTGGATGACAAAGC | 379 | 57 |
| | Aac(3)-II-R | AGGCGACTTCACCGTTTCTT | | |

Aac(3)-II) for 30 s; extension at 72 °C for 30 s; and a final extension at 72 °C for 7 min. The S-PCR products were verified using gel electrophoresis on a 2% agarose gel and stained with GelStain (Beijing Transgen Biotech Co., Ltd., Beijing, China).

## Construction and verification of positive plasmids

The positive plasmids with resistance genes were constructed as described in a previous study (Li et al., 2019). In brief, the DNA fragments of the target genes were obtained by PCR reactions using the genomic DNA of *E. coli* as a template. The target fragment was inserted into the pMD 19-T simple vector and transformed into *JM109*-competent cells. The positive clones were selected for overnight culture to extract plasmids. Finally, the copies of recombinant plasmids were calculated.

## Sensitivity and accuracy evaluation of S-PCR

After the positive plasmids were constructed, the sensitivity of S-PCR reactions was evaluated using the serially diluted 10-fold positive plasmids. According to the resistance gene information of 237 clinical strains of *E. coli* provided by the First People's Hospital of Yunnan Province, the strains with *Aac(6′)-Ib*, *Ant(3″)-Ia*, *Aph(3′)-Ia*, and *Aac(3)-II* resistance genes were screened to evaluate accuracy.

## Establishment of M-PCR reaction system and accuracy evaluation of M-PCR

The M-PCR was performed by using a Multiplex PCR kit (Nanjing Vazyme Biotechnology Co., Ltd., Nanjing, China). According to the manufacturer's protocol, the M-PCR reaction system containing 25 μL of 2X Multiplex Buffer, 10 μL of 5X Multiplex GC Enhancer, 1 μL of each primer (10 μM), 1 μL of Multiplex DNA polymerase, and 1 μg of DNA template from each strain, was added with nuclease-free water up to 50 μL. The reactions were performed in a GeneAmp PCR System 9700 (Thermo Fisher Scientific, Inc., Waltham, MA, USA) with the following amplification conditions: pre-denaturation at 95 °C for 5 min, followed by 30 cycles of denaturation at 95 °C for 30 s, annealing at 60 °C for 3 min, extension at 72 °C for 3 min, and a final extension at 72 °C for 30 min.

The M-PCR products were verified by gel electrophoresis on 2% agarose gel and stained with GelStain (Beijing Transgen Biotech Co., Ltd., Beijing, China).

Based on the drug sensitivity information of 237 clinical strains of *E. coli*, five isolates (*1611NY0004*, *1611UR0282*, *1611SP0549*, *1611UR0062*, and *1611UR0215*) previously tested for the *Aac(6′)-Ib*, *Ant(3″)-Ia*, *Aph(3′)-Ia*, and *Aac(3)-II* by S-PCR were tested for the M-PCR.

### Sensitivity evaluation of M-PCR

The sensitivity of M-PCR was performed by using gradient dilution plasmids and bacterial solution. The four equal concentration plasmids with *Aac(6′)-Ib*, *Ant(3″)-Ia*, *Aph(3′)-Ia*, and *Aac(3)-II* resistance genes were mixed together and serially diluted to 10-fold ($10^8$–$10^0$). *E. coli 1611NY0004* was selected as the representative strain for sensitivity evaluation. The stain was cultured in LB liquid medium to $OD_{600} = 1$, and the colony-forming units of bacterial solution were calculated by the plate count method. The bacterial solution was serially diluted as 10-fold ($10^8$–$10^0$).

### M-PCR detection for clinical samples

All the 237 clinical *E. coli* strains were provided and identified by the First People's Hospital of Yunnan Province and used in the accuracy evaluation of S-PCR and M-PCR.

## RESULTS

### Establishment of S-PCR reaction system

The positive plasmids with drug resistance genes (*Aac(6′)-Ib*, *Ant(3″)-Ia*, *Aph(3′)-Ia*, and *Aac(3)-II*) were successfully constructed, with $3 \times 10^9$, $4 \times 10^9$, $5 \times 10^9$ and $4 \times 10^{10}$ copies/µL (114.5, 141.2, 156.8 and 153.2 ng/µL), respectively. The S-PCR reaction template comprised the serially diluted plasmids. Meanwhile, the four drug resistance genes were detected in the 237 clinical strains of *E. coli*, and the bacterial solution was the S-PCR template. As shown in Fig. 1, the limitation for the detection of the four drug resistance genes was $10^0$ copies/µL. The accuracy rate for the detection of drug resistance genes was 100% (partially representative results), which was consistent with the hospital data (Table S1).

### Establishment of M-PCR reaction system

After the S-PCR reaction system was established and optimized (data not shown), the M-PCR reaction system was constructed. As shown in Fig. 2, *Aac(6′)-Ib*, *Ant(3″)-Ia*, *Aph(3′)-Ia*, and *Aac(3)-II* with 112, 180, 317, and 379 bp, respectively, were amplified successfully. *E. coli 1611NY0004* with the four drug resistance genes was used as the positive control template for the M-PCR reaction. In the M-PCR reaction system, the four drug resistance genes could be amplified well simultaneously.

### Accuracy evaluation of M-PCR

The five clinical *E. coli* strains (*1611NY0004*, *1611UR0282*, *1611SP0549*, *1611UR0062*, and *1611UR0215*) with four resistance genes were screened for the accuracy evaluation of M-PCR. The four equal concentration plasmids with *Aac(6′)-Ib*, *Ant(3″)-Ia*, *Aph(3′)-Ia*,

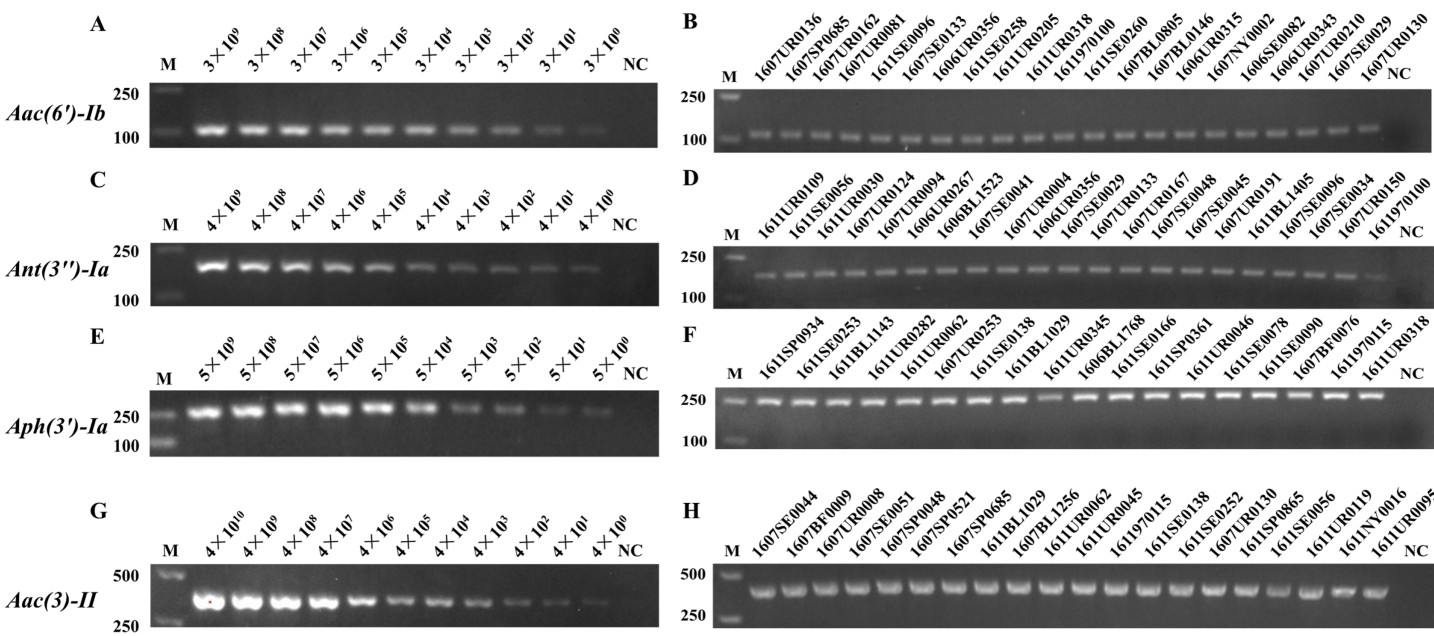

**Figure 1** **The sensitivity and accuracy evaluation of *Aac(6′)-Ib*, *Ant(3″)-Ia*, *Aph(3′)-Ia*, and *Aac(3)-II* resistance genes by S-PCR.** The serially diluted positive plasmids with resistance gene were used as the template in the sensitivity evaluation of S-PCR. (A) *Aac(6′)-Ib*, $3 \times 10^9$–$3 \times 10^0$ copies/µL. (C) *Ant(3″)-Ia*, $4 \times 10^9$–$4 \times 10^0$ copies/µL. (E) *Aph(3′)-Ia*, $5 \times 10^9$–$5 \times 10^0$ copies/µL. (G) *Aac(3)-II*, $4 \times 10^{10}$–$4 \times 10^0$ copies/µL. Meanwhile, the four resistance genes were detected in the 237 clinical strains of *E. coli* respectively, the bacterial solution as the S-PCR template. (B) The accuracy evaluation of *Aac(6′)-Ib* resistance genes. (D) The accuracy evaluation of *Ant(3″)-Ia* resistance genes. (F) The accuracy evaluation of *Aph(3′)-Ia* resistance genes. (H) The accuracy evaluation of *Aac(3)-II* resistance genes. All experiments were repeated six times. The nuclease-free water used as template for NC (Negative control). M, marker.

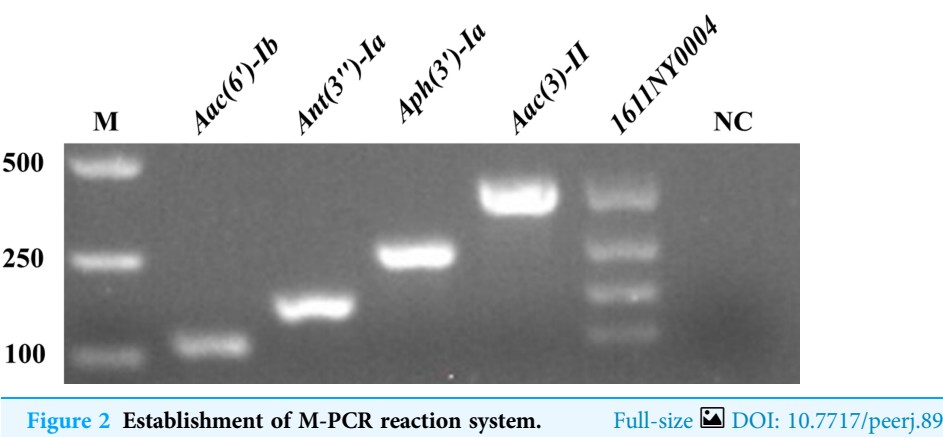

**Figure 2** **Establishment of M-PCR reaction system.**

and *Aac(3)-II* resistance genes were mixed together as the positive control. As shown in Fig. 3, the four resistance genes were successfully amplified by M-PCR in the tested strains, which was consistent with the resistance gene information provided by the hospital.

## Sensitivity evaluation of M-PCR

The gradient dilution of plasmids and bacterial solution was used in the sensitivity evaluation of M-PCR. The gradient dilution of plasmids was the mixture of the four equal

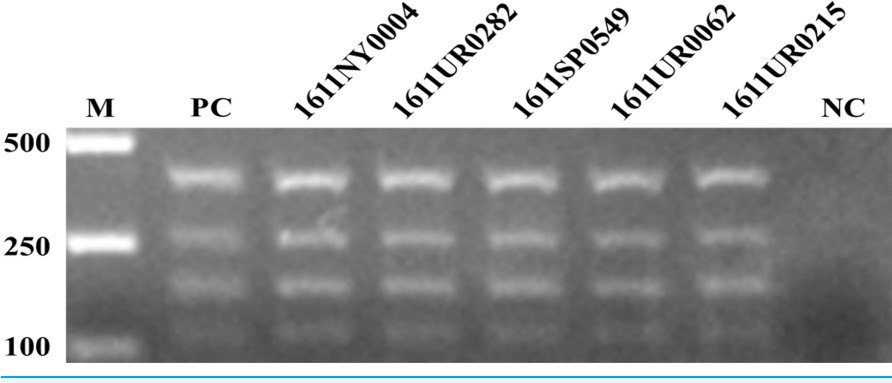

**Figure 3 The accuracy evaluation of M-PCR.**

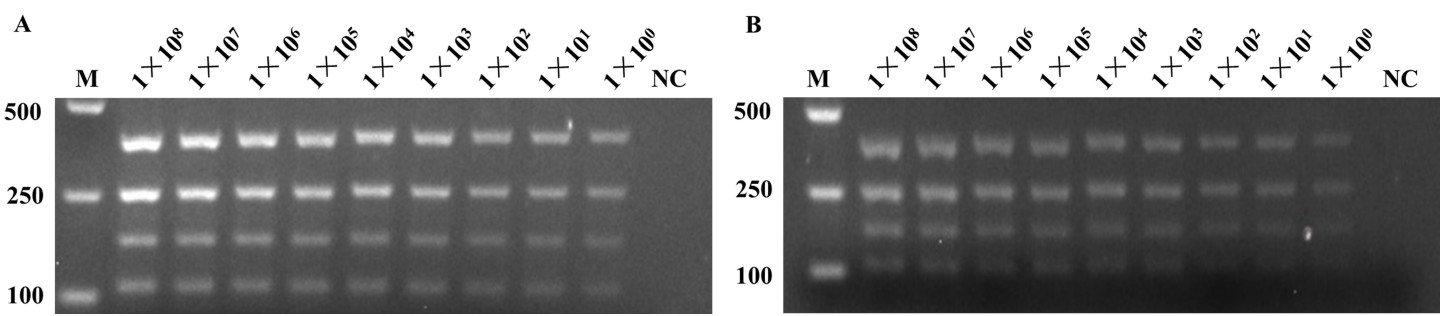

**Figure 4 The sensitivity evaluation of M-PCR.** The gradient dilution of plasmids and bacterial solution was used in the sensitivity evaluation of M-PCR. (A) The four equal concentration plasmids with *Aac(6′)-Ib*, *Ant(3″)-Ia*, *Aph(3′)-Ia*, and *Aac(3)-II* resistance genes were mixed, and serially diluted as 10-fold ($1 \times 10^8 - 1 \times 10^0$), as the sensitivity evaluation of M-PCR. The nuclease-free water used in lane 10 as template for NC. (B) The *E. coli 1611NY0004* strain with four resistance genes was used as the template for the the sensitivity evaluation of M-PCR. The bacterial solution was serially diluted as 10-fold ($1 \times 10^8 - 1 \times 10^0$). The nuclease-free water used in lane 10 as template for NC. NC, negative control; M, marker.

concentration plasmids with *Aac(6′)-Ib*, *Ant(3″)-Ia*, *Aph(3′)-Ia*, and *Aac(3)-II* resistance gene and then serially diluted as 10-fold ($10^8 - 10^0$ copies/mL). The *E. coli 1611NY0004* strain was used in the gradient 10-fold ($10^8 - 10^0$ CFU/mL) dilution of bacterial solution for sensitivity evaluation. As shown in Fig. 4, the detection system could reach the detection limit of $10^0$ copies/mL or $10^0$ CFU/mL at the plasmid level and bacterial liquid level.

### Detection of clinical samples by multiplex PCR

All the 237 *E. coli* strains were detected by S-PCR and M-PCR. The detection results and the accuracy are shown in Table S1. The results showed that the detection rates of *Aac(6′)-Ib*, *Aac(3)-II*, *Ant(3″)-Ia*, and *Aph(3′)-Ia* were 32.7%, 59.2%, 23.5% and 16.8% in 237 clinical strains of *E. coli*, respectively. These values were consistent with the gene information of the strains given by the hospital.

## DISCUSSION

Antibiotics are considered the most effective drugs in the treatment of infectious diseases (*Elder, Kuentz & Holm, 2016*). The emergence of antibiotics changed the outcome of

infectious diseases and extended life expectancy (*Wagner & Maynard, 2018*). However, given the overuse and misuse of antibiotics, MDR strains have emerged (*Elder, Kuentz & Holm, 2016*). Therefore, rapid and accurate methods for the detection of bacterial resistance are urgently needed, and the use of antibiotics or antibiotic therapy should be more standardized and technically, where possible, monitored. With the development of molecular biology and bioinformatics, the detection methods of bacterial drug resistance genes, including PCR, loop-mediated isothermal amplification, and whole genome sequencing (WGS) (*Moran et al., 2017*; *Su, Satola & Read, 2019*), have increased (*Tamburro & Ripabelli, 2017*). Although WGS is gradually reaching maturity, while sequencing costs are drastically decreasing, it is however still relatively costly, complex in operation and requires for analysis specialized personnel like bioinformaticians, which limits its popularization and application (*Quainoo et al., 2017*). M-PCR is widely used in the identification and drug resistance detection of bacteria for its rapid, sensitive, economy-friendly and high-effect characteristics (*Pham et al., 2017*).

Aminoglycoside antibiotics are one of the most widely used antibiotics, and the main reason for aminoglycoside antibiotic resistance is the production of AME. Aminoglycoside antibiotic resistance is related to the AME drug resistance genes *Aac(6′)-Ib*, *Aac(3)-II*, *Ant(3″)-Ia*, and *Aph(3′)-Ia* (*Zarate et al., 2018*). Thus, this study designed primers for the four drug resistance genes and constructed a four-drug resistance genes detection system, which greatly improved the detection efficiency and shortened the detection time.

The MDR gene detection system was established successfully in this experiment, and it exhibited the advantages of high efficiency, rapidity, and high performance-to-price ratio compared with the S-PCR reaction system. The MDR gene detection system also fully met the requirements for the clinical detection of pathogens and drug resistance. In this experiment, the four drug-resistant genes were identified in the five representative *E. coli* MDR bacteria by M-PCR. The results obtained by M-PCR were consistent with those of S-PCR, thereby indicating the accuracy of M-PCR. The M-PCR detection system was applied to the detection of 237 clinical strains of *E. coli*. On the basis of the drug resistance gene information given by the hospital, the accuracy rate of M-PCR could reach 100%; the detection rates of *Aac(6′)-Ib*, *Aac(3)-II*, *Ant(3″)-Ia*, and *Aph(3′)-Ia* were 32.7%, 59.2%, 23.5% and 16.8%, respectively. These findings indicated the potential of reintroducing additional resistant genes into the M-PCR system to detect numerous genes at a time. Even the specific genes of the bacteria could be added into the M-PCR system for the identification of bacterial species. From the detection rate of the four aminoglycoside modifying enzyme resistance genes, *Aac(3)-II* had the highest detection rate in the 237 clinical strains of *E. coli*. This result suggested that the resistance of three aminoglycoside antibiotics gentamicin, tobramycin, and netilmicin resistance is widespread in bacterial infections and should be avoided.

## CONCLUSION

The M-PCR system developed in this study could amplify the four genes with very extremely high sensitivity to $10^0$ copies. The sensitivity rate of M-PCR was higher than that of most previously reported studies, which proved the sensitivity of this

technique. Moreover, the M-PCR detection method greatly reduced the detection time and improved the detection efficiency. M-PCR in this study demonstrated high sensitivity and efficiency, low price, and other characteristics but not one-to-one correspondence between genotype and phenotype. Therefore, simply detecting genes failed to completely determine the phenotype, but the four genes selected genes had the highest prevalence of aminoglycoside antibiotic resistance. We conducted the simultaneous detection of these four genes to compensate for the limitations of detecting one gene, thereby making our methodology reliable and persuasive. Our method was established to assist medical institutions to predict drug resistance, to provide an accurate direction for clinical drug selection, and to meet the needs of both doctors and patients for rapid diagnosis. In the future, we will propose a rapid drug sensitivity identification technique based on phenotype. We will also combine traditional detection methods with modern molecular techniques, starting from the phenotype, for the accurate identification of bacterial drug resistance.

## ACKNOWLEDGEMENTS

We are grateful for the strains donated and the drug resistant information provided by the First People's Hospital of Yunnan Province.

### Funding

This study was supported by the Yunnan Science and Technology Commission (grant numbers: 2015BC001, 2015DH010). The funders had no role in study design, data collection and analysis, decision to publish, or preparation of the manuscript.

### Grant Disclosures

The following grant information was disclosed by the authors:
Yunnan Science and Technology Commission: 2015BC001, 2015DH010.

### Competing Interests

The authors declare that they have no competing interests.

### Author Contributions

- Yaoqiang Shi performed the experiments, prepared figures and/or tables, and approved the final draft.
- Chao Li performed the experiments, prepared figures and/or tables, and approved the final draft.
- Guangying Yang analyzed the data, prepared figures and/or tables, and approved the final draft.
- Xueshan Xia analyzed the data, authored or reviewed drafts of the paper, and approved the final draft.
- Xiaoqin Mao analyzed the data, authored or reviewed drafts of the paper, and approved the final draft.

- Yue Fang analyzed the data, authored or reviewed drafts of the paper, and approved the final draft.
- A-Mei Zhang conceived and designed the experiments, authored or reviewed drafts of the paper, and approved the final draft.
- Yuzhu Song conceived and designed the experiments, authored or reviewed drafts of the paper, and approved the final draft.

## Data Availability

Raw data is available in Table S1.

## Supplemental Information

Supplemental information for this article can be found online at http://dx.doi.org/10.7717/peerj.8944#supplemental-information.

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
