# Peer review of "A rapid and accurate method for the detection of four aminoglycoside modifying enzyme drug resistance gene in clinical strains of Escherichia coli by a multiplex polymerase chain reaction"

_PeerJ, doi:10.7717/peerj.8944_

## Round 0.1 · original submission · Major Revisions

Your paper raised a long series of scientific questions. Moreover, both reviewers complained about the poor language. Taken as a whole, your paper should be rejected. However, both reviewers also found merits in your studies. Thus, I'd like to give you the opportunity to revise your paper if you are willing (and able) to make the very *major* changes that are requested. If you do so, please (i) prepare a detailed rebuttal in which you explain, point by point, how you have dealt with the scientific remarks raised by both reviewers; (ii) prepare a new version of the paper with all significant changes highlighted so that we can quickly see which are the changes and where they are in your new text; (iii) have both the rebuttal and the new version carefully checked by an English-speaking colleague knowledgeable in the field of study (this is very important as PeerJ does not provide copy-editing service). Be also aware that your new version will undergo a new round of review by the same or by different reviewers. I cannot, therefore, make any commitment at this stage about a final acceptance of your submission. The way you revise it and how convincing the new version is will be key elements in the final decision.

Reviewer 1 ·

Basic reporting

The manuscript requires significant editing by a native English speaker. As written, it is ambiguous and difficult to read. I have attempted to edit the abstract and introduction as an example of the extensive work that needs to be done.

Experimental design

The multiplex PCR strategy is reasonable. The data presented demonstrate that the approach is sounds and can identify four aminoglycoside resistance elements. What is not clear is why these particular genes were selected and wether this information will alter clinical practice as suggested in the abstract and introduction. Nevertheless, as a standard method to detect these genes, the work is sound.

Validity of the findings

No comment

Additional comments

The manuscript needs to be rewritten with improved English language use and clear reasons why these gens in particular were elected and how the information can be used. Technically, the work is sound.

Annotated reviews are not available for download in order to protect the identity of reviewers who chose to remain anonymous.

Reviewer 2 ·

Basic reporting

See below

Experimental design

See below

Validity of the findings

See below

Additional comments

This article by Yaoqiang Shi et al. and entitled “ A rapid and accurate method for the detection of four aminoglycoside modifying enzymes drug resistance genes in clinical strains of Escherichia coli by the multiplex polymerase chain reaction ” is interesting and fits well with the idea outlined in the introduction and discussion, namely that a quick identification of the involved pathogen(s) and their drug susceptibility pattern is essential for a good antibiotic use.

Although already several groups published papers in this sense the article under focus here has it’s merit but, honestly, I think that it is not ready for publishing. I propose a major revision.

Main comments:

English style and/or phrasing is often not qualitatively good for the standard of the journal as well as for the scientific community. It is not specifically the wording or style but the fact that this could give misunderstandings or confusions. Therefore I propose that a native English speaking scientist helps the team to adjust.

I suggest an edit to the title:

Title:
A rapid and accurate method for the detection of four aminoglycoside modifying enzymes drug resistance genes in clinical strains of Escherichia coli by the multiplex polymerase chain reaction:

suggestion:
A rapid and accurate method for the detection of four aminoglycoside modifying enzyme drug resistance genes in clinical strains of Escherichia coli by a multiplex polymerase chain reaction

I suggest edits to the Abstract:

Background. Antibiotics were and are (were the most) very effective drugs in the treatment of infectious diseases. Aminoglycoside antibiotics were one of the most common antibiotics in the treatment of bacterial infections, while the development (or emergence) of drug resistance against those medicines (of these antibiotics were very) serious (or maybe better significant).

AIM: develop an efficient, rapid, accurate and sensitive detection method, applicable in routine clinical use

Methods. (To establish an efficient, rapid, accurate, sensitive, and multiple drug
resistance detection method for clinical Escherichia coli.)
Escherichia coli (E. coli) was taken as model organism for the development of a rapid, accurate, and reliable multiplex polymerase chain reaction (M-PCR) (system) for the detection (identification) of (the) four aminoglycoside modifying enzyme(s) (AMEs) resistance genes Aac(6')-I, Aac(3')-II, Ant(3'), and Aph(3') (was established).

(Also,) The M-PCR was used to detect the distribution of AMEs resistance genes in 236 clinical strains of E. coli. (And) The results were verified by the
simplex polymerase chain reaction (S-PCR).

Results. The results of M-PCR and S-PCR
showed that the detection rate of Aac(6')-I, Aac(3')-II, Ant(3'), and Aph(3') were 32.6%, 59.3%, 23.7%, and 16.9% in 236 clinical strains of E. coli respectively. Compared with the traditional methods for the detection and identification, (a) the rapid and accurate M-PCR detection method was established for the detection of AMEs drug resistance genes, which was significant to guide the reasonable(y) use of drugs, and prevent the spread of bacterial resistance.


Comments on the abstract:
Already in the abstract I find that for example the conclusion is certainly not clear in the sense that it says that the method developed “was significant to guide “reasonably” the antibiotherapy and prevent the spread of bacterial resistance…I find it to vague, also further in the paper which is technically and scientifically not at the required level of the journal.

Scientifically I find that there is too much room for subjectivity. In order to be accurate and specific I propose to define what is meant by Multi Drug Resistance (MDR) since I have the impression MDR, here, is not well defined and could lead to confusions. A good paper related to this issue is the one of Magoriakos et al. (in Clin Microbiol Infect 2012:18:268-281) “ Multidrug-resistant, extensively drug-resistant and pandrug-resistant bacteria: an international expert proposal for interim standard definitions for acquired resistance” in which guidelines for MDR and definitions are discussed and proposed.

Since we only are concerned about aminoglycosides it would be useful to add following review by Krause et al. “Aminoglycosides: an overview” (Cold Spring Harbor Perspectives in Medicine, 2016; available at www.perspectivesinmedicine.org).

I would also suggest to shorten the paper by cutting in the introduction as well as in the discussion where we find several redundancies.

This will make the article much better and focussed without losing the importance of the message that the need is great for the development and daily use of more rapid pathogen identification and drug susceptibility tests in routine medical settings. In that frame this work has to be evaluated and every contribution into well described and evaluated technical approaches, especially the identification and validation of ‘good targets’ (genes in casu is) necessary and useful. This statement is still through although a lot of similar technical research and even commercially available assays is conducted and exist. An interesting and inspiring article related to this is the one of Chavada and Maley in the Open Microbiology Journal (2015, 9, 125-135) entitled “Evaluation of a Commercial Multiplex PCR for Rapid Detection of Multidrug Resistant Gram Negative Infections”.

No in-depth validation was performed and should be performed. Only 5 “MDR” were tested on the background of xxx E. coli strains. Comments? Check!

Was it the aim only to target E. coli ? Why not other Enterobacteriaceae where the same ABRs are found? Could this also not be extrapolated to some non-fermenters like P. aeruginosa

Shortcomings of genome technologies, in casu PCR-technology: could not it be that a gene is present but not well expressed, meaning that the phenotype should give a certain susceptibility?

A recent paper amongst others is the one of Moran et al. (in J Antimicrob Chemother 2017; 72:700-704) “Prediction of antibiotic resistance from antibiotic resistance genes detected in antibiotic-resistant commensal Escherichia coli using PCR or WGS”. This aspect is not discussed neither just mentioned. I think this issue needs a certain attention in the discussion.

Another interesting paper is the article of Su et al. in J Clin Microbiol (2019, 57: 1405-1418) entitled “Genome-based prediction of antibiotic resistance”.

Both articles are very interesting and could be taken up in the global discussion as well as in the references.

The use of “etc. should absolutely be avoided in a scientific and technical paper! It occurs several times.

Also I would suggest that for example more rigour should be applied in for example:

See introduction line 49: …were mainly used to treat infections caused by facultative aerobic gram-negative bacteria, such as Escherichia coli, Klebsiella pneumoniae, and non-fermenters like Pseudomonas aeruginosa, etc 15,22,41.

Line 56: … there is an urgent need …

---

## Round 0.2 · Major Revisions

As mentioned in my first communication to you, your revised version was submitted to the original reviewers for comments. One of them, while being globally happy with the improvements made, still suggested a series of important corrections. If you are ready to do the work needed and further improve your submission, I'd gladly examine it again. Please, bear in mind that this should be the very final version. So, pay attention to all remarks and critiques and make the necessary changes. If you disagree with some of these remarks, explain why. Your rebuttal will be an essential element for me to reach a final decision. Since I will need to look at your new version and your rebuttal, I cannot make any commitment at this stage about a final acceptance of your work.
I look forward seeing your new version and rebuttal in due course.

Reviewer 2 ·

Basic reporting

.

Experimental design

.

Validity of the findings

.

Additional comments

Title OK
Line 37-39:

This technique can be used for the clinical detection as well as the surveillance and monitoring of the spread of those specific antibiotic resistance genes.

Line 47:
I would delete …”and super bacteria.” Since I find it redundant with “pan-drug-resistant”

37-51: OK with addition of the Magiorakis definitions.

Line 52: Addition of Krausse et al. OK

Line 53: please correct the typing error in the word “Gram”
Line 56: please correct However, bacterial can… into However, bacteria can…
Line 59: add into the sentence: … tolerance has become a global worldwide concern.
Line 93: I would change into … to timely guide the clinical antibiotic therapy.

Line 102:
Please change into or rephrase as suggested:
Given that the M-PCR can detect multiple genes simultaneously we aimed to develop a M-PCR system for the detection of the four most widely spread AME genes. The reaction system was verified in 237 clinical E. coli strains.
Lines 151 and further:

I would suggest to rephrase it as follows if I understood well the meaning:
Based on the drug sensitivity information of 237 clinical strains of E. coli, five isolates (…) previously tested for the Aac(6’)-Ib, Ant(3”)-Ia, Aph(3’)-Ia and Aac(3)-II by S-PCR were each tested for the M-PCR.

Line 164: 236 or 237?

Line 177: I think the …(data unknown)… means and should be …(data not shown)…?

Line 210:
I suggest to rephrase into
…, and the use of antibiotics or antibiotic therapy should be more standardized and technically, where possible, monitored.

Line 213:
Although WGS is gradually reaching maturity, while sequencing costs are drastically decreasing, it is however still relatively costly, complex in operation and requires for analysis specialised personnel like bioinformaticians

---

## Round 0.3 · accepted · Accept

Thank you making the necessary corrections suggested by the reviewer. The paper is now stronger and more readable. I guess it will be of interest to many clinical microbiologists when facing the need to characterize the mechanisms of resistance to aminoglycosides.